# Bio-Assisted Synthesis and Characterization of Zinc Oxide Nanoparticles from *Lepidium sativum* and Their Potent Antioxidant, Antibacterial and Anticancer Activities

**DOI:** 10.3390/biom12060855

**Published:** 2022-06-20

**Authors:** Bisma Meer, Anisa Andleeb, Junaid Iqbal, Hajra Ashraf, Kushif Meer, Joham Sarfraz Ali, Samantha Drouet, Sumaira Anjum, Azra Mehmood, Taimoor Khan, Mohammad Ali, Christophe Hano, Bilal Haider Abbasi

**Affiliations:** 1Department of Biotechnology, Quaid-i-Azam University, Islamabad 45320, Pakistan; bismameer786@gmail.com (B.M.); ansaandleeb097@gmail.com (A.A.); junaidbiotech34@gmail.com (J.I.); hajraashraf67@gmail.com (H.A.); jsa_09@yahoo.com (J.S.A.); taimoor@unizwa.edu.om (T.K.); 2Institute of Chemistry, University of the Punjab, Lahore 54590, Pakistan; kushifmeer@gmail.com; 3Laboratoire de Biologie des Ligneux et des Grandes Cultures (LBLGC), Plant Lignans Team, UPRES EA 1207, INRA USC1328, Université d’Orléans, F 28000 Chartres, France; samantha.drouet@univ-orleans.fr; 4Department of Biotechnology, Kinnaird College for Women, Lahore 54000, Pakistan; sumaira.anjum@kinnaird.edu.pk; 5Stem Cell & Regenerative Medicine Lab, National Centre of Excellence in Molecular Biology, University of Punjab, 87-West Canal Bank Road, Lahore 53700, Pakistan; azramehmood@cemb.edu.pk; 6Natural and Medical Sciences Research Center, University of Nizwa, Sultanate 616, Oman; 7Centre for Biotechnology and Microbiology, University of Swat, Charbagh 01923, Pakistan; alimoh@uswat.edu.pk

**Keywords:** bio-assisted synthesis, ZnO NPs, phytochemicals, antioxidant activity, cytotoxicity, anticancer activity

## Abstract

Nanotechnology is an emerging area of research that deals with the production, manipulation, and application of nanoscale materials. Bio-assisted synthesis is of particular interest nowadays, to overcome the limitations associated with the physical and chemical means. The aim of this study was to synthesize ZnO nanoparticles (NPs) for the first time, utilizing the seed extract of *Lepidium sativum.* The synthesized NPs were confirmed through various spectroscopy and imagining techniques, such as XRD, FTIR, HPLC, and SEM. The characterized NPs were then examined for various in vitro biological assays. Crystalline, hexagonal-structured NPs with an average particle size of 25.6 nm were obtained. Biosynthesized ZnO NPs exhibited potent antioxidant activities, effective α-amylase inhibition, moderate urease inhibition (56%), high lipase-inhibition (71%) activities, moderate cytotoxic potential, and significant antibacterial activity. Gene expression of caspase in HepG2 cells was enhanced along with elevated production of ROS/RNS, while membrane integrity was disturbed upon the exposure of NPs. Overall results indicated that bio-assisted ZnO NPs exhibit excellent biological potential and could be exploited for future biomedical applications. particularly in antimicrobial and cancer therapeutics. Moreover, this is the first comprehensive study on *Lepidium sativum*-mediated synthesis of ZnO nanoparticles and evaluation of their biological activities.

## 1. Introduction

Nanotechnology has been recently exploited as a tool to investigate unilluminated approaches in many ways, such as artificial implants, imaging, targeted drug delivery, sensing and gene-delivery systems [1]. Generally, nanoparticles [2] constitute 20–15,000 atoms, their size is smaller than 100 nanometers, and they exist in a realm that bestrides the Newtonian and quantum scales. Nanoparticles can be generated from various materials in numerous shapes, such as wires, spheres, tubes, and rods [3,4]. Biological synthesis of nanoparticles has several benefits over chemical synthesis, including economic feasibility, simplicity, being free of harmful waste materials, being ecofriendly, and the availability of plant metabolites as capping and stabilizing agents [5]. NPs derived from plants are more diverse and show consistency in size and shape, in contrast to those synthesized by other organisms [6].

Zinc oxide (ZnO) has fascinated several researchers from different disciplines of sciences due to its wide applications and unique characteristics [7]. ZnO is referred as a wide-band-gap semiconductor (3.36 eV and has possessed great consideration in striking electronic applications because of its unique chemical, optical, and electrical features [8]. Nanostructure availability at a wide range results in ZnO being an ideal material for biotechnology, for nanoscale optoelectronics, and for piezoelectric nanogenerators. The ZnO NPs size, surface area, crystallinity, and band gap are strongly linked with activities, i.e., high conductivity [9], good electric and thermal ability, significant antimicrobial potential, long stability, and higher photocatalytic properties [10]. ZnO NPs exhibits strong resistance against microorganisms [11]. Studies revealed that CaO, ZnO, and MgO possessed strong antibacterial activity [12], which is linked with reactive-oxygen-species formation on the surface of oxides and is studied by the conductometric method [11], as shown in Figure 1. It is also employed as an effective drug-delivery system [13].

The benefit of utilizing these inorganic oxides as antimicrobial agents is that they constitute the mineral elements necessary to humans, and they possess strong activity when intaken in a minute quantity [14]. ZnO NPs prepared by conventional techniques, such as laser ablation [15], the sol–gel method, solvothermal, chemical route [16], the microwave method [17,18], and inert gas condensation. Such methods needs high pressure, inert gases such as helium, toxic chemicals, and laser radiations, so are costly in contrast to a green synthesis process [19]. There is a desperate requirement to produce easy, less expensive, simple to manage, and ecofriendly methods for the preparation of NPs that can lower the utilization of hazardous chemicals [20]. The characteristics and yield of prepared ZnO NPs are strongly associated with reaction parameters such as temperature, pH, and time. A wide range of ZnO NPs synthesized from seeds have been reported in the literature, such as the stem of *Boswellia ovalifoliolata* [21], peel of *Citrus sinensis* (orange) [22], *Silybum marianum* (L.) [23], fruit of *Rubus coreanus* [24], etc. Nowadays, algae [25], yeast [26], fungi, bacteria [27], wild extracts of plants [28], and in vitro-derived callus and plant extracts [29] are utilized to produce nanoparticles. It is also suggested that plant-mediated synthesis of ZnO NPs, when applied on the HepG2 cell line, induce apoptosis by increasing the level of ROS/RNS that results in the disruption of mitochondrial-membrane integrity, and biocompatibility analysis on hRBCs showed that ZnO NPs are slightly hemolytic in nature [16].

The current study aimed to synthesize first-time bio-assisted ZnO NPs from the seed extract of *Lepidium sativum.* Generally, *Salmonella typhi* showed resistant against antibiotics, but in the current study significant antibacterial activity was examined in contrast to most potent antibiotic “cefexime”. The genus Lepidium belongs to the family of Brassicaceae: it is basically an edible herb with length of approximately 50 cm. It originated from Southwest Asia and Egypt, but now it is cultivated throughout the world. It is utilized widely as an anti-spasmodic, antioxidant, analgesic hepatoprotective, galactagogue, anti-inflammatory, diuretic, anti-diarrheal, etc. [30]. Phytochemical studies revealed that *L. sativum* constitutes sterols, volatile oil, alkaloids, carotene, fixed oil, and glycosides. In seeds of *L. sativum* alkaloids, dimeric Lepidine B, C, D, E, and F as well as semilepidinoside A and B were found. Sinapin as well as sinapic acid were also obtained from the methanolic extract of defatted seeds [31]. Characterization of these ZnO NPs was completed by XRD, FTIR, HPLC, and SEM. Moreover, this study focused on various biological assays, constituting antioxidant, enzyme-inhibition assays, catalytic activity, cell viability, brine-shrimp-lethality assays, biocompatibility assays, membrane-integrity analysis, caspase activity, and antibacterial assays, which were conducted to investigate the potency of bio-assisted NPs.

## 2. Materials and Methods

### 2.1. Chemicals

All the reagents utilized in the experimentation were from Sigma-Aldrich and Merck (Saint Quentin Fallavier, France).

### 2.2. Seed Collection and Preparation of Seed Extract

The *Lepidium sativum* seeds used in this study were collected from Gujranwala District, Punjab, Pakistan. The wild seeds were taxonomically identified at the Plant Sciences Department, Quaid-i-Azam University Islamabad, Pakistan. Aqueous extract (1:10) was prepared by the addition of 30 g of plant-seed powder into 500 mL flasks constituting 300 mL of distilled water. The flasks were placed over a magnetic stirrer for 2 h at 80 °C. The extract was filtered twice with a nylon cloth for the removal of solid plant residues, followed by filtration three times by utilizing Whatman filter paper to exclude any remaining residues. Processing of the filtrate was completed for further utilization.

### 2.3. Bio-Assisted Synthesis of ZnO Nanoparticles

Bio-assisted ZnO NPs were prepared by the procedure followed by Abbasi et al. (2017). Briefly, 50 mL of 0.02 M zinc acetate dihydrate solution was prepared and kept in the stirrer for 2 h at 60 °C. After this, 1 mL plant extract was added to the solution, followed by constant stirring under the continuous dropwise incorporation of 2 M NaOH, until the pH of the solution was maintained at 12. The solution was kept overnight. After the appearance of white precipitants, the solution was allowed to settle down the dissolved precipitate by centrifugation for 10 min at 10,000 rpm. The supernatant was discarded, and the pellet was washed three times with distilled water. The precipitates were re-dissolved in distilled water, poured in the clean petri-plate and kept in an incubator for drying at 60 °C for 24 h. The dried nanoparticles were grinded to obtain fine powder of ZnO nanoparticles and were used for further characterizations and prediction of biological efficacy. Flow chart of the complete study design is shown in Figure 2.

### 2.4. Characterization

Various characterization techniques were performed to predict structural, functional and morphological characteristics along with identification of phytochemicals in bio-assisted ZnO NPs.

#### 2.4.1. X-ray Diffraction (XRD) Analysis

X-ray Diffractometer (AXS DS Advance, Bruker, Billerica, MA, USA) was exploited to predict crystalline nature of bio-assisted ZnO NPs. XRD instrument has a cathode ray emitting X-rays on samples through which X-ray diffraction analysis (XRD) performed. Composition of zinc oxide NPs was evaluated by powder XRD in the 2θ region, from 0° to 80°. The average particle size of ZnO NPs was obtained by Scherrer equation. The equation is given as follows:(1)$$D=\frac{\kappa \lambda }{\beta \cos\theta }$$
where, k = Shape Factor (0.94), λ = X-rays wavelength (1.5421Å), β = Full width at half maximum in radians, θ = Bragg’s Angle.

#### 2.4.2. Fourier Transform Infrared Radiation Spectroscopy (FTIR) Analysis

Fourier Transform Infrared Radiation Spectroscopy (FTIR, Bruker, Billerica, MA, USA) was conducted in the spectral array from 400 to 4000 cm^−1^ for the prediction of major functional groups that may act as a capping, reducing or stabilizing agent during bio-assisted synthesis of ZnO NPs.

#### 2.4.3. High Performance Liquid Chromatography (HPLC) Analysis

The HPLC (Merck, Saint Quentin Fallavier, France) analysis was completed by the procedure followed by [32]. 5 µm particle size was formed by 250 × 4.6 mm, Hypersil PEP 300 C18, 10 × 4.1 mm and guard column Alltech was utilized for the purpose of separation at 35 °C. The compounds were analyzed at a wavelength of 320 nm and 520 nm. Mobile phase constituted two HPLC grade solvents i.e., A = HCOOH/H_2_O, pH = 2.1 and B = CH_3_OH. Composition of the mobile phase alter throughout 1 h each run, with non-linear gradients as follows: 8% B, 100% B, 33% B, 30% B, 12% B, and 8% B for 36 min, 30–35 min, 28 min, 17 min, 11 min and 0 min at 1 mL/min flow rate. Among each individual run about 10 min re-equilibration time was utilized and quantification was conducted. Experiment was run thrice and results were presented as µg/mg DW of samples.

#### 2.4.4. Scanning-Electron-Microscopy (SEM) Analysis

Size as well as morphology of prepared ZnO NPs were predicted by utilizing scanning-electron microscope (SEM, Jeol JSM-6510LV). Preparation of samples were completed by placing nanoparticles (10 µL) on a cover slip for each micrograph followed by placing overnight to dry. Then, the samples were evaluated with SEM.

### 2.5. In Vitro Biological Activities of ZnO NPs

Various biological assays were performed to evaluate bio-assisted ZnO NPs potency.

#### 2.5.1. Antioxidant Assays

The total antioxidant capacity (TAC) of prepared ZnO NPs was determined by a phosphomolybdenum-based assay utilizing the method of [33]. Initially 100 µL of each fraction, refraction (4 mg/mL extract in DMSO), and positive control (ascorbic acid, 1 mg/mL) was mixed with reagent (900 µL) constituting 0.6 M sulphuric acid, 28 mM sodium phosphate, and 4 mM ammonium molybdate. The reaction mixture was kept in a water bath at 95 °C for 90 min, the cooling of the test samples was completed at room temperature, and 200 µL of this reaction mixture was shifted to a 96-well plate. Absorbance was recorded at 630 nm, and the results were presented as µg AAE/mg.

The total-reducing potential (TRP) of the test samples were predicted by utilizing the methodology of [33]. Briefly, from a 4 mg/mL extract prepared in DMSO, about 200 μL of extract was mixed with 400 μL of phosphate buffer (0.2 mol/L, pH 6.6) and 1% potassium ferricyanide [K_3_Fe (CN)_6_], which was then incubated for 20 min at 50 °C. After this, 400 μL of 10% trichloroacetic acid was mixed to the mixture, and centrifugation was completed at 3000 rpm for 10 min by a centrifuge (Spectrafuge™ 24D microcentrifuge, Labnet International, Corning, NY, USA). The 200 μL from the upper layer solution was added into a 96-well plate, and, then, to stop the reaction, 50 μL of 0.1% Ferric chloride was added. Finally, at 630 nm the absorbance was recorded, and the results were presented as µg AAE/mg.

The antioxidant capability of the test samples was evaluated by utilizing the stable 2, 2-diphenyl 1-picrylhydrazyl (DPPH) free radical as described by [34]. Spectrophotometric analysis was conducted to analyze the percentage of radical-scavenging activity (%RSA) in terms of % inhibition. Briefly, in a 96-well plate, 10 μL extract was transferred, followed by a 190 μL addition of DPPH solution, and the plate was incubated for 30 min at 37 °C. Assay was conducted in triplicate, and ascorbic acid was used as a standard. The following equation was used to calculate scavenging activity as a percentage (%inhibition):(2)$$\%\;\mathrm{Scavenging}\;=\left(1-\frac{\mathrm{Abs}}{\mathrm{Abc}}\right)*100$$
where Abs indicates the DPPH-solution absorbance with the sample, and Abc shows the negative-control (constituting the reagent devoid of the sample) absorbance.

#### 2.5.2. Enzyme-Inhibition Assays

α-Amylase-Inhibition Assay

The α-Amylase-inhibition capability of ZnO NPs was determined by a protocol followed by [35]. Initially, in a 96-well plate, about 15 µL phosphate buffer (pH 6.8) was poured, 0.14 U/mL α-amylase enzyme solution was prepared, and 25 µL of it was poured into another 96-well plate, followed by the subsequent mixing of 10 µL of the test sample (4 mg in DMSO) and 40 µL of a starch solution (2 mg/mL in potassium-phosphate buffer). Incubation of the test samples was performed for 30 min at 50 °C, followed by the addition of 1 M HCl (20 µL) and 90 µL iodine reagent (5 mM iodine, 5 mM potassium iodide). The 100% enzyme activity represented by the negative control did not constitute any text sample, while acarbose, at a concentration range of 5–200 µg, was utilized as a positive control. A blank was prepared devoid of amylase enzyme and the test sample. Absorbance was recorded utilizing a microplate reader (Thermo Scientific Multiskan GO) at 540 nm, and the percentage of inhibition is measured as follows:(3)$$\%\;\mathrm{enzyme}\;\mathrm{inhibition}=\mathrm{OD}\left(s\right)-\frac{\mathrm{OD}\left(n\right)}{\mathrm{OD}\left(b\right)}*100$$
where OD (s) = absorbance value of test sample, OD (n) = absorbance of negative control, and OD (b) = absorbance of blank.

Lipase-Inhibition Assay

The lipase-inhibition assay previously reported by [36] was followed with some minor modifications. In ultra-pure water (10 mg/mL), lipase was dissolved and the supernatant was utilized followed by centrifugation via centrifuge (Spectrafuge™ 24D microcentrifuge, Labnet International, Corning, NY, USA) at 16,000 rpm for 5 min. Tris Buffer (100 mM; pH 8.2) was taken as an assay buffer, and olive oil was utilized as a substrate. It was prepared by 0.08% *v*/*v*, dissolved in 5 mM sodium acetate (pH 5.0), containing 1% Triton X-100 heated in boiling water for 1 min to aid dissolution, mixed well, and cooled for further use at room temperature. Each Eppendorf for the test sample constituted 350 µL of buffer, 150 µL of lipase, and 50 µL of the test sample (4 mg/mL in DMSO), along with the addition of 450 µL of substrate to start the reaction. An Eppendorf devoid of the test sample was pondered as a blank, and Orlistat was used as an inhibitor and constituted 400 µL of buffer, 150 µL of lipase, and 450 µL of substrate. Incubation of all the samples was performed at 37 °C for 2 h, followed by centrifugation at 16,000 rpm for 1 min. After centrifugation, 200 µL of mixture was poured into the respective wells of a microtiter plate, and absorbance was taken at 400 nm via a UV spectrophotometer (Halo DR-20, UV-VIS spectrophotometer, Dynamica Ltd., Victoria, Australia). Comparison with a standard inhibitor (Orlistat) was completed, and the percentage of enzyme inhibition was determined by the following formula:(4)$$\%\;\mathrm{enzyme}\;\mathrm{inhibition}=\mathrm{OD}\left(b\right)-\frac{\mathrm{OD}\left(s\right)}{\mathrm{OD}\left(b\right)}*100$$
where OD (b) = absorbance of blank, and OD (s) = absorbance value of test sample.

Urease-Inhibition Assay

The assay mixture containing 25 µL of urease, 50 µL of phosphate buffer ((3 Mm, pH 4.5) constituting 100 mM urea), and 10 µL of the test samples (4 mg in DMSO) was added in a 96-well plate and kept in an incubator at 30 °C for 15 min. Later, 45 µL of phenol reagent (1% (*w*/*v*) phenol and 0.005% (*w*/*v*) sodium nitroprusside) and 70 µL of alkali reagent (0.5% (*w*/*v*) NaOH and 0.1% NaOCl) were mixed with each well. The activity of urease inhibition was evaluated by predicting the production of ammonia that was apparent, with the pungent smell of ammonia as discussed by [37]. Incubation of the plates was completed at 30 °C for 50 min and later a 630 nm reading was taken by utilizing a UV spectrophotometer (Halo DR-20, UV-VIS spectrophotometer, Dynamica Ltd., Victoria, Australia). Thiourea was utilized as a urease inhibitor and considered as a control, while a blank constitute with none of the test sample and control was utilized, which contained 60 µL of buffer instead of 50 µL, thoughthe rest was the same as above. The percentage of enzyme inhibition was determined by the help of the following formula:(5)$$\%\;\mathrm{enzyme}\;\mathrm{inhibition}=\mathrm{OD}\left(b\right)-\frac{\mathrm{OD}\left(s\right)}{\mathrm{OD}\left(b\right)}*100$$
where OD (b) indicates blank, and OD (s) indicates value of the sample.

#### 2.5.3. Catalytic Activity

Peroxidase (POD) Activity

The peroxidase activity of the test samples was investigated by utilizing the method followed by [38], with minor modifications. For the assay, in each well 140 μL of NaAc-HAc buffer (0.2 M, pH 4.0) was poured, followed by the addition of 20 μL of the test sample, and later H_2_O_2_ (6 mM, freshly prepared) and 20 μL TMB (3 mM, freshly prepared) were poured. A reaction mixture devoid of the test sample was considered as a control. Absorbance was measured by utilizing a micro-plate reader at 652 nm wavelength, and enzyme activity was determined by the following formula:(6)A=ELC

Here, A: sample absorbance, C: enzyme concentration (mM/min/mg), E: extinction coefficient (6.39 mM^−1^ cm^−1^), and L: length of wall (0.25 cm).

#### 2.5.4. Cytotoxicity Assays

Cell-Viability Assay (XTT Assay)

The cytotoxicity of the biosynthesized ZnO NPs was checked against NIH3T3 mouse fibroblast cells by XTT assay (2, 3-bis [2-methoxy-4-nitro-5- sulfoxyphenyl]-2H-tetrazolium 5-carboxyanilide inner salt), utilizing an XTT assay kit as previously elaborated by [39]. Briefly, fibroblast cells were trypsinized using 1X Trypsin + EDTA and plated in a 96-well plate at a cell density of 3000 cells/well, provided with the complete growth medium of Dulbecco’s Modified Eagle Medium Low Glucose (DMEM LG) (Sigma Aldrich, St. Louis, MO, USA), along with 20% fetal bovine serum (FBS) (Gibco), and incubated in a 5% CO_2_ incubator at 37 °C. At 80%–90% confluency, the cells’ media was aspirated, and washing of the cells was completed twice with 1x PBS. The cells were treated with different concentration of ZnO NPs (25, 50, 75, and 100 μg/mL) prepared in serum-free media, and the plates were incubated for 24 h at 37 °C. Untreated fibroblast cells were considered as a control. After 24 h of nanoparticle treatment, the media was aspirated from the wells and washed twice with 1xPBS. After this, a fresh mixture of electron-coupling reagents and XTT was prepared in a ratio of 1:50, and 50 μL of the prepared mixture was added in each treated well of the plate. The plate was then wrapped completely in aluminum foil and incubated in a humidified 5% CO_2_ incubator at 37 °C for 4 h. The absorbance was recorded at 450 nm, and the experiment was conducted thrice in a triplicate manner.

Evaluation of Anticancer Potential by MTT Assay

The anticancer potential of the prepared ZnO NPs was assessed against HepG2 liver-cancer cells. MTT (3-(4,5-dimethylthiazolyl-2)-2,5-diphenyltetrazolium bromide), referred to as a tetrazolium dye, was utilized to determine the in vitro-toxicity impact of extracts/NPs, as followed by [23]. In living cells, a reduction in MTT occurred in an insoluble formazan (purple product) that is measured spectrophotometrically. For 24 h, pre-seeded HepG2 cells (>90% viability; 1 × 10^4^ cells/well; 200 µL per well) in a 96-well plate were exposed with 200 µg/mL of the test samples. It was followed by the addition of 10 µL of MTT dye (5 mg/mL) per well, along with 3 h of incubation. Then, insoluble formazan was dissolved by the addition of 10% acidified sodium dodecyl sulfate (SDS), and incubation of the cells was completed overnight. By utilizing a microplate reader (Thermo Scientific Multiskan GO), analysis of each plate was completed at 570 nm. Untreated HepG2 cells (NTC) acted as a control. Prior to cytotoxicity screening, centrifugation of the extract was performed, and the sonication of NPs was completed by ultrasonic bath (USC1200TH, Prolabo, Sion, Switzerland). The percentage (%) of cell viability in contrast to the NTC sample was quantified, utilizing the following equation:(7)$$\%\;viability=\left(Sample\;Abs-Control\;Abs/NTC’\;s\;abs-Media\;Abs\right)*100$$
where the Abs of the NTC represents optical density at 570 nm, respectively, for the non-treated control samples, while the Abs of the sample corresponds to the treated control samples. The Abs of the sample control, and the Abs of blank corresponds to the background optical density. The whole experiment was repeated thrice.

Brine-Shrimp-Lethality Assay

Bio-assisted synthesized ZnO NPs (20 mg/mL stock in water) were used to determine the lethality against *Artemia salina* (brine shrimp) in the 96-well plate (300 µL) for about 24 h. The brine shrimp is significant in the investigation of the toxicological impact of nanoparticles or other compounds. *Artemia salina’s* larvae was utilized in this study by following the protocol of [23]. Brine-shrimp eggs were subjected to incubation of 24–48 h for hatching in seawater. During the procedure, the constant supply of oxygen in the sterile sea water (38 g/L) was ensured, and supplementation of 6 mg/L dried yeast was given with the sterile sea water under the proper light. The necessary temperature (30–32 °C) and light for hatching were provided by illumination. By utilizing a Pasteur pipette, 10 mature phototropic nauplii were picked and added into the wells. After this, 200 µg/mL final concentrations of ZnO NPs were added into the wells containing the sea water and shrimp larvae. In each well, a final volume of 300 µL was adjusted. For the positive control, doxorubicin’s serial concentration (ranging from 1 µg/mL to 10 µg/mL) were taken, while 1% DMSO in sea water served as a negative control. Quantification of the live shrimps was completed after 24 h of incubation with nanoparticles, and the median lethal concentration (LC50) was calculated by utilizing a 2D v5.01 table curve of the test NPs, with a mortality rate of ≥50%.

Biocompatibility Testing with Human Red Blood Cells (hRBCs) (Hemolytic Assay)

Hemolytic assay was conducted to investigate the biocompatibility of bio-assisted ZnO NPs against freshly isolated human red blood cells [23]. Fresh blood was collected with consent from 1 female and 2 male healthy students (average age 28 years), having no previous illness record. The blood was dispensed in an EDTA tube to prevent clotting. For the isolation of red blood cells, the centrifugation of 1 mL blood was completed for 5 min at 14,000 rpm, and the obtained pellet was washed with PBS twice. In 200 µL of pelleted erythrocyte, 9.8 mL of PBS (phosphate-buffer saline) (pH: 7.2) was added and mixed thoroughly. Almost 100 µL of erythrocyte suspension and the test NPs sample were introduced into a 1.5 mL Eppendorf tube. The tubes were incubated at 35 °C for 1 h, proceeded by centrifugation at 10,000 rpm for 10 min. Then, 100 µL of supernatant was dispensed in a 96-well plate, and absorbance of the released hemoglobin was recorded at 540 nm utilizing an Absorbance Microplate Reader (Thermo Scientific Multiskan GO). Triton X-100 served as a positive control, while DMSO acted as a negative control. The results were presented in the form of the percentage of hemolysis, using the following formula:(8)$$\begin{array}{cc} \%\;\mathrm{hemolysis} & =[\mathrm{Sample}\;\mathrm{Abs}\\ & -\mathrm{Negative}\;\mathrm{control}\;\mathrm{Abs}/\;\mathrm{Positive}\;\mathrm{control}\;\mathrm{Abs}\\ & -\mathrm{Negative}\;\mathrm{control}\;\mathrm{Abs}] \end{array}$$

#### 2.5.5. Membrane-Integrity Analysis

Measurement of Reactive Oxygen and Nitrogen Species

Dihydrorhodamine-123 (DHR-123) fluorescent dye was utilized to measure the reactive oxygen species (ROS)- and reactive nitrogen species (RNS)-level, as followed by [40]. In the presence of ROS and RNS, the dihydrorhodamine-123 (DHR-123) dye become oxidized into fluorescent rhodamine (R123). Briefly, HepG_2_ cells at a cell density of 5 × 10^5^ cells/well were plated, and after 90% confluency the cells were treated with ZnO NPs or DMSO (control group). After the NPs treatment, the treated and non-treated cells (NTC) were washed twice with PBS, resuspended in PBS containing 0.4 μM of DHR-123, and incubated in the dark at 30 °C for 10 min. After washing twice with PBS, the fluorescence signal was measured at 505 nm with an emission wavelength of 535 nm, respectively. The assay was repeated twice using Resveratrol as a positive control, and the results were expressed in term of Trolox C equivalent antioxidant capacity (TEAC).

Evaluation of Mitochondria-Membrane Potential

The mitochondria-membrane potential (ΔΨm) was determined by investigating the fluorescence of 3,3′-dihexyloxacarbocyanine iodide DiOC6, which is a specific probe, as followed by [41]. On the base of ΔΨm, DiOC6 stains the mitochondria. For this purpose, ZnO NPs were treated, while NTC HepG_2_ cells were grown in a culture medium containing 25 nM of DiOC6 and were incubated at 30 °C for 45 min. By utilizing a Versa Fluor Fluorimeter, the fluorescence signal was measured at 482 nm. For each condition, six independent measurements were conducted, and the results were represented as relative fluorescent units.

Caspase-3/7-Like Activities

The protein lysates from the bio-assisted synthesized ZnO NPs that treated HepG2 cells and control cells were isolated in a cold lysis buffer (1 mM DTT, protease inhibitors in PBS and 1% NP40). With the help of SDS-PAGE, a total of 50 mg of separated proteins were tracked by immunoblotting, using specific primary antibodies for caspase-3 and caspase-7 with dilution of 1:1000 using an ECL identification kit [42].

#### 2.5.6. Antibacterial Assay

The antibacterial activity of the ZnO NPs was examined by the disc-diffusion method, as illustrated by [35]. For this purpose, two Gram-positive bacteria (*Micrococcus luteus* and *Staphlococcus aureus)* and three Gram-negative bacteria (*Salmonella typhi*, *Enterobacter aerogens and Salmonella Setubal)* were tested. The bacteria were grown on nutrient agar plates, and 5 μL (4 mg/mL in DMSO) of the test samples were impregnated on filter paper discs and placed in the inoculated plates. Cefexime was used as a positive control, and the plates were incubated at 37 °C for 24 h. After overnight incubation, the average diameter of the clear zone of inhibition was measured and recorded.

#### 2.5.7. Statistical Analysis

Origin 8.5 (Windows v8.1, Northampton, MA, USA) was used for the result analysis of all performed activities, while the statistical analysis of the XTT cell-viability assay was performed by a one-way analysis of variance (ANOVA) test, followed by an unpaired Bonferroni test, using GraphPad Prism 8 software. Data were represented as mean ± SD of three independent experiments, followed by a one-way ANOVA (*p* < 0.05).

## 3. Results

### 3.1. Bio-Assisted Synthesis of Zinc-Oxide NPs

The medicinal plant Lepidium sativum was exploited for the successful bio-assisted synthesis of ZnO NPs for the very first time. The white-crystalline powder of the ZnO NPs was acquired at pH 12, after several steps of washing, drying, and grinding. It was taken and stored in an air-tight glass vial, at room temperature for physiochemical, morphological, and biological activities.

### 3.2. Physical Characterization

#### 3.2.1. XRD (X-ray Diffraction) Analysis

The purity, phase identification, and structure of the bio-assisted ZnO NPs were predicted by X-ray diffraction. The crystalline nature and purity of NPs was confirmed by the XRD pattern, with several diffraction peaks predicted at different 2θ, i.e., 31.74°, 34.34°, 36.35°, 47.38°, 56.57°, 62.68°, 66.36°, 67.83°, and 68.2°, corresponding to different Miller indices (100), (002), (101), (102), (110), (103), (200), (212), and (201), respectively, as shown in Figure 3a. The average particle size of the pure ZnO NPs was evaluated to be 25.6 nm.

#### 3.2.2. Fourier Transform Infrared Spectroscopy (FTIR)

FTIR spectroscopy was utilized to predict the surface adsorption of the functional groups present on the bio-assisted ZnO NPs. FTIR spectra of the bio-assisted ZnO NPs is in spectral range of 400–4000 cm^−1^, as shown in Figure 3b. The absorption peaks were observed in the region of 617 cm^−1^, 882 cm^−1^, 1225 cm^−1^, 1286 cm^−1^, 1470 cm^−1^, 2370 cm^−1^, 2920 cm^−1^, 2980 cm^−1^, 3647 cm^−1^, 3811 cm^−1^, 3892 cm^−1^, 3933 cm^−1^, and 3983 cm^−1^, respectively. A characteristic band predicted at 617 cm^−1^ corresponds to the Zn-O stretching bond, as ZnO NPs were reported in the region of 650–400 cm^−1^ [5]. The peak intensity at 882 represents the C-H “oop” of the aromatics, 1225 and 1286 correspond to the C-N stretch of aliphatic and aromatic amines, 1470 corresponds to the Amine NH vibration stretch, as previously indicated by [43], 2370 corresponds to C≡N stretching mode, and 2920 and 2980 correspond to the C-H stretch of alkanes, respectively. The peak at 3377 corresponds to the O-H group [44]. The bands of absorption observed in the region of 3600–3900 cm^−1^ correspond to the stretching-vibration modes of the OH groups [45]. These results demonstrate the significant importance of biological molecules in ZnO NPs fabrication.

#### 3.2.3. HPLC Analysis

HPLC analysis showed various secondary metabolites associated with bio-assisted ZnO NPs, such as chlorogenic acid (830 µg/mg DW), quercetin (1850 µg/mg DW), and kaempferol (1290 µg/mg DW). Chlorogenic acid belongs to the phenolic family, while quercetin and kaempferol belong to the flavonoid group. The chemical structures of chlorogenic acid, quercetin, and kaempferol are shown in Figure 4.

#### 3.2.4. Scanning-Electron-Microscopy (SEM) Analysis

Particle size, along with surface morphology of the bio-assisted ZnO NPs, was estimated utilizing a scanning-electron microscope (SEM). Representative images of the scanning-electron micrograph of the bio-assisted nanoparticles are shown in Figure 5. A typical scanning-electron micrograph reveals that particles possess spherical shape with some degree of aggregation.

### 3.3. In Vitro Antioxidant Potential

Total antioxidant capacity (TAC), total reducing power (TRP), and DPPH-free radical scavenging activity assays were conducted to evaluate the antioxidant capability of the bio-assisted ZnO NPs and phytochemicals as shown in Figure 6a. For the assays, 4 mg/mL concentration were exploited. The TAC value was found to be 96.60387 ± 0.57116 µgAAE/mg for the plant extract and 97.75848 ± 0.91892 µgAAE/mg for the NPs. The TRP assay was also conducted to further evaluate the antioxidant potency of the plant extract and the NPs. The TRP value for the plant extract and the NPs was predicted to be 68.4898 ± 0.68483 µgAAE/mg and 73.03813 ± 0.78838 µgAAE/mg, respectively. To further evaluate the antioxidant potential of the plant extract and the bio-assisted ZnO NPs, DPPH radical scavenging assay was conducted. Results of the plant extract and the NPs were found to be 55.7% and 61.033%, respectively. From the results, it can be revealed that some of the compounds of the aqueous extract of Lepidium sativum were responsible for the stabilization as well as the reduction in the ZnO NPs during the synthesis of the NPs, while the seed extract itself also possessed good antioxidant potential.

### 3.4. Enzyme-Inhibition Activities

Enzyme-inhibition activities were shown in Figure 6b. The α-amylase-inhibition capability of the *L. sativum* extract and the ZnO NPs were found to be 12.8% and 16.3% inhibition, respectively. The α-amylase inhibition assay demonstrated a moderate anti-diabetic activity for the biosynthesized ZnO NPs and extract, while there is no significant difference observed between the percentage of inhibitory potential of the ZnO NPs and the extract against α-amylase enzyme.

Lipase-inhibition-assay results indicated that the *L. sativum* extract and ZnO NPs exhibited 68.7% and 71.8% lipase-inhibition capacity. The lipase-inhibition assay performed in this study exhibited good lipase-inhibitory potential for both the biosynthesized ZnO NPs and the extract, however there is no significant difference between the percentage of inhibition of the NPs and the extract. So, these results demonstrate that both the *L. sativum* extract and the ZnO NPs can act as lipase inhibitors.

The urease-inhibition potential of the *L. sativum* extract and the ZnO NPs were observed to be 76.1% and 57.0%. According to the results, both the ZnO NPs and the extract showed good inhibitory properties against the urease enzyme, however the percentage of inhibitory potential of the extract is significantly higher as compared to that of the NPs.

### 3.5. Catalytic Activity

Peroxidase activity (POD) was conducted using a 4 mg/mL concentration of the ZnO NPs and 1 ml of the *L. sativum* extract. Results showed that plant extract and NPs exhibited peroxidase activity of 0.2 ± 0.01 mM/min/mg and 0.4 ± 0.01 mM/min/mg, respectively, as shown in Figure 6c. The results elaborated on the greater catalytic potential of the ZnO NPs compared to the *L. sativum* extract, suggesting an improved sensitivity of H_2_O_2_ for the nanoparticles than the plant extract.

### 3.6. Cell-Viability Assay

#### 3.6.1. XTT Assay

NIH3T3 fibroblast cell lines were used to determine the cytotoxic impact of the bio-assisted ZnO NPs. Different concentrations of the ZnO NPs were tested, and the results revealed the dose-dependent cytotoxicity of the NPs, i.e., they were less toxic at lower doses and more toxic at higher concentrations. XTT-assay results showed 100.0 ± 0.012% cell viability in the control group vs. 75.23 ± 1.866% in 25 µg/mL, 71.10 ± 1.784% in 50 µg/mL, 55.63 ± 1.468% in 75 µg/mL, and 52.13 ± 1.64% in 100 ug/mL, respectively, as shown in Figure 7a. The 25 ug/mL concentration of the NPs offered less toxicity to the cells in contrast to the control and the other tested concentrations of the NPs (50 µg/mL, 75 µg/mL, and 100 µg/mL). However, there is no significant difference observed between cell viabilities at 25 ug/mL and 50 ug/mL concentrations, and a similar trend was observed for 75 ug/mL and 100 ug/mL concentrations of the NPs. The 100 µg concentration exhibited the least cell viability and is the most toxic among all the tested concentrations, as it reduces the overall viability of the cells to 50%. Hence, these results conferred that a dose above 100 µg/mL may appear to be lethal to the cells.

#### 3.6.2. Antiproliferative Potential of ZnO NPs by MTT Assay

The antiproliferative activity of the bio-assisted ZnO NPs (20 mg/mL) against the HepG2 cancer cell line was tested using an MTT cell-viability assay. Results depicted that the bio-assisted ZnO NPs showed cytotoxicity towards HepG2 cells. Non-treated cells (NTCs) showed a percentage of viability of 95 ± 1.71%, which reduced to 30.10 ± 1.34% in the presence of the bio-assisted ZnO NPs at a dose of 200 µg/mL, as shown in Figure 7b. These results revealed a higher antiproliferative effect by the ZnO NPs at 200 µg/mL concentration against HepG2 cells, hence confirming the significant anticancer ability of the ZnO NPs against liver-cancer cells.

#### 3.6.3. Evaluation of Toxicity by Brine-Shrimp-Lethality Assay

The obtained results indicated that significant toxicity was shown by the bio-assisted ZnO NPs against brine shrimp larvae. Doxorubicin presented a 5.92 ± 0.34 µg/mL LC50 value, while the ZnO NPs presented an LC50 value of 19.43 ± 1.90 µg/mL, respectively, as shown in Table 1. Results of the brine shrimp were stated in different standards as follows: if LC50 < 1.0 µg/mL, then the compounds are highly toxic; compounds are said to be toxic if LC50 is 1.0–10.0 µg/mL; for moderately toxic compounds, LC50 is 10.0–30.0 µg/mL; the LC50 value for mildly toxic compounds is 30.0–100.0 µg/mL, and non-toxic compounds exhibit an LC50 > 100.0 µg/mL [46,47]. Hence, the bio-assisted ZnO NPs were considered as moderately toxic, while doxorubicin showed more toxicity compared to the NPs, as it is a toxic compound and has been used to cure various cancers.

#### 3.6.4. Biocompatibility Analysis with Human Red Blood Cells (hRBCs)

The bio-safe nature of the bio-assisted ZnO NPs was evaluated by assessing their compatibility with human red blood cells (hRBCs). Results revealed that the bio-assisted ZnO NPs showed 4.1 ± 0.2% hemolytic potency, compared to the NTCs that exhibited 0.9 ± 0.3% hemolytic ability, as shown in Table 1. The hemolysis potential is determined by the rupturing of the RBC and the release of hemoglobin upon treatment of the 4 mg/mL ZnO NPs. The findings of this assay suggested slightly hemolytic potency of the ZnO NPs, compared to the control.

### 3.7. Membrane-Integrity Analysis

#### 3.7.1. Reactive Oxygen- and Nitrogen-Species Assessment

The results depicted that the ZnO NPs accelerate the level of the ROS and RNS in HepG2 cells, in contrast to non-treated cells (NTCs). The least ROS/RNS production was observed in the case of the NTCs, i.e., 835 ± 80.17, while the bio-assisted ZnO NPs exhibited a high level of ROS/RNS production, i.e., 3009.67 ± 401.48, as shown in Figure 7a. This is suggesting that membrane integrity is disrupted by the elevated level of the reactive oxygen and nitrogen species.

#### 3.7.2. Evaluation of Mitochondria-Membrane Potential

Results showed that a loss of mitochondrial function was observed in the case of the bio-assisted ZnO NPs, as shown in Figure 8a. The ZnO NPs presented 1393.7 ± 56.2, while non-treated cells (NTCs) showed mitochondrial-membrane potential to be 3374.9 ± 105.3.

#### 3.7.3. Gene Expression of Caspases 3/7

The protein expression of caspase-3 and caspase-7 was determined in response to the bio-assisted ZnO NPs showing that the bio-assisted ZnO NPs elevated the caspase-3/7 protein activity of, i.e., 224.0 ± 11.1%, and 337.7 ± 16.8% for the ZnO NPs, while the non-treated cells (NTCs) exhibited protein activity that was 100.0 ± 1.8% and 100.0 ± 6.6%, as shown in Figure 8b.

### 3.8. Antibacterial Activity

The antibacterial potency of the bio-assisted ZnO NPs was tested against five pathogenic bacterial strain, among these were two Gram-positive bacteria (*Micrococcus luteus* and *Staphlococcus aureus)* and three Gram-negative bacteria (*Salmonella typhi, Enterobacter aerogens* and *Salmonella Setubal),* using the well-diffusion method. A 4 mg concentration of NPs and 1 mL of pure plant extract were used to evaluate bacterial susceptibility. A sample exhibiting a ≥12 mm inhibition zone is considered significant. Results indicated that the plant extract did not exhibit anti-bacterial activity, as its zone of inhibition is less than 12 mm, while the ZnO NPs exhibited good anti-antibacterial potential against all bacterial strains, although *Salmonella Setubal* and *Staphlococcus aureus* antibacterial activity were found to be the most significant. The inhibitory zones were measured in millimeters, with the help of Vernier caliper. The inhibitory zones of the NPs obtained at a 4 mg/mL concentration were recorded as 18 ± 1.1 mm for *Staphlococcus aureus*, 18 ± 1.4 mm for *Salmonella Setubal*, 15 ± 1.2 mm for *Micrococcus luteus*, 15 ± 1.2 mm for *Salmonella typhi*, and 14 ± 1 mm for *Enterobacter aerogens*, respectively, as shown in Figure 9. While the inhibitory zones measured against cefexime were 19 ± 1.8 mm for *Staphlococcus aureus*, 22.5 ± 2.2 mm for *Salmonella Setubal*, 21.8 ± 0.8 mm for *Micrococcus luteus*, 13 ± 2.5 mm for *Salmonella typhi*, and 20 ± 2.2 mm for *Enterobacter aerogens*, respectively.

## 4. Discussion

In this study, an aqueous seed extract of *Lepidium sativum* was exploited as a reducing and stabilizing agent for preparation of the ZnO NPs for various biological activities. *Lepidium sativum*, also commonly known as garden cress or pepper cress, belongs to the *Brassicaceae* family and is an edible herbaceous plant [48,49]. Different plant parts, particularly the seeds, exhibit potent pharmaceutical characteristics, and these seeds have been utilized previously to cure cough, bronchitis, and asthma. Cress seeds have been utilized to cure leucorrhea and hemorrhoids and have been proven to be effective against skin illnesses and diarrhea [50,51,52]. Due to its high phytochemical profile, the dietary utilization of this medicinal plant increases the body natural immunity against several diseases. These phytochemicals, such as carotenoids, flavonoids, phenolics, and terpenoids, have a potent role in the protection of cells from oxidative stress, which is responsible for some metabolic disorders such as cancer [53,54]. *L. sativum* constitutes a large amount of the phenolic components that can serve as antioxidant and anticancer agents [55]. Several phytochemicals constituted by *L. sativum* includes ferulic acid, vanillic acid chlorogenic acid, kaempferol, *p-*coumaric acid, caffeic acid, and quercetin. These phytochemicals could have performed a potent function for the preparation of stable bio-assisted ZnO NPs [48]. After synthesis of such stable bio-assisted ZnO NPs, in the form of white precipitates, they were characterized by X-ray diffraction (XRD), Fourier transform infrared spectroscopy (FTIR), HPLC, and scanning-electron microscopy (SEM).

X-ray-diffraction analysis revealed the purity and phase identification of the bio-assisted ZnO NPs. The peaks predicted at different 2θ verified the pure and high crystalline nature of the particles. By studying the literature, it is revealed that the hexagonal wurtzite structure (JCPDF file NO. 00-036-1451) [56] of the particles was confirmed by Miller indexation. Similar results regarding size and shape were also reported in previous studies [57,58,59,60]. Moreover, the average particle size of the bio-assisted ZnO NPs was found to be 25.6 nm, using Scherrer formula. FTIR analysis exhibited the functional groups that existed in the plant extract and may have participated in the mechanism of bonding with the bio-assisted ZnO NPs. As *L. sativum* extract contains saponosides, alkaloids, terpenes, flavonoids, sterols, and tannins compounds [61], and these phytochemicals can act as a reducing and capping agent during nanoparticle synthesis [62,63], it was demonstrated previously that the functional groups in the plant extract donate electrons that are responsible for zinc-ion reduction (Zn^2+^ to Zn^1+^),which finally reduces to zinc NPs (Zn^0^). Another study indicated that the negative functional groups present in the extract exhibited a stabilizing effect [64]. A typical scanning-electron micrograph reveals that particles possess a spherical shape with some degree of aggregation. Similar morphological studies were also found in previous studies [65,66].

In vitro biological assays were also conducted to determine the biomedical applications of the ZnO NPs. The TAC and TRP assays involve an investigation of the reductones present in the samples. The reductones are referred to as species with antioxidant potential, due to their capability to donate an H-atom, which leads to the discontinuation of free-radical chains [67]. A DPPH assay depends on the production of the light-yellow diphenyl picrylhydrazine molecule, developed due to a reduction in DPPH moiety, after electron acceptance from the donor species [68]. The antioxidant potential of the bio-assisted ZnO NPs was found to be slightly greater in contrast to that of the plant extract. Related observations from the literature review suggested the high antioxidant potential of the bio-assisted NPs [69,70]. The overall results obtained from these activities revealed that the compounds present in the aqueous extract of *L. sativum* may have been involved in the stabilization and reduction in the ZnO NPs during the synthesis process, which, hence, resulted in the relatable, good antioxidant and reduction potential of the NPs, as exhibited by the extract.

Amylase inhibitors, also termed as starch blockers, contain components that can stop the absorption of dietary starches by the body, through hindering the degradation of complex sugars into their simpler forms. Such materials possess potent applications to control diabetes [23]. This significant enzyme breaks down the α bonds of polysaccharides, i.e., glycogen as well as starch, yielding maltose, and glucose [71]. In the light of the obtained results, it can be concluded that the *L. sativum* extract and the ZnO NPs can act as mild anti-diabetic agents. Previous reports in this context indicated that the polyphenols and flavonoids capping the NPs could be responsible for the α-amylase inhibition of bio-assisted NPs [72,73].

A lipase-inhibition assay was performed to identify the anti-obesity properties of the *L. sativum* extract and the bio-assisted ZnO NPs. This fat digestion is decreased or inhibited by suppressing the pancreatic-lipase action that splits triglycerides into fatty acids and glycerol. Orlistat, which is a potent lipase-inhibitor drug has been used to treat obesity and it was used as a positive control in this assay [36,74].As no significant difference was noticed between the lipase inhibition of the NPs and the extract, these results demonstrate that both *L. sativum* extract and the ZnO NPs can act as lipase inhibitors. It has been revealed from previous reports that naturally occurring polyphenols have the ability to inhibit pancreatic lipase, and, hence, in this way, they influence energy intake and affect fat digestion [75]. Hence, the obtained results are in accordance with previous work.

Urease (E.C 3.5.1.5), the first enzyme crystallized from *Canavalia ensiformis*, is known to possess nickel ions that can rapidly catalyze the breakdown of urea to generate carbon dioxide and ammonia [37]. The higher inhibitory potential of the extract in the results may be attributed to the greater count of the phytochemicals present in the extract that may cause urease inhibition, compared to the phytochemical count that is coated on the NPs. It has been demonstrated earlier that the functional groups and chemical constituents of phytochemicals play a significant role in urease inhibition. The functional groups, such as ketones and the hydroxyl group, which are associated with aromatic rings, can show the interaction with the active site of enzymes with Ni ions, resulting in impeding the function of urease [37].

Catalytic-peroxidase activity determines the ability of the oxidoreductase that catalyzes the H_2_O_2_ breakdown, by donating an electron and by the oxidation of the inorganic and organic compounds [76]. The results showed the greater catalytic potential of the ZnO NPs in contrast to the *L. sativum* extract, suggesting an improved sensitivity of H_2_O_2_ for the nanoparticles. The same trend is reported by [77], which is that ZnS-MMT nanocomposites improved H_2_O_2_ sensitivity.

NIH3T3 fibroblast cell lines were exploited to predict the cell viability of the bio-assisted ZnO NPs. So, the results demonstrated that the 25 ug/mL concentration of NPs possessed less toxicity and more cell viability, while the 100 µg/mL concentration exhibited the least cell viability and more toxicity. Earlier studies also revealed that the ZnO NPs possessed greater cell viability at a lower concentration, though it depends on exposure time and concentration because they have a direct impact on cell viability [78]. Moreover, a higher antiproliferative effect of the synthesized ZnO NPs at a 200 µg/mL concentration against HepG2 cells showed the potent anticancer ability of the ZnO NPs against liver-cancer cells. In previous reports, the ZnO NPs have shown an efficient antiproliferative effect against liver-cancer cells [23]. Our findings are also relevant to previous work by [28]. Brine shrimp at the larval stage were used to determine the toxicological impact of ZnO NPs. Brine shrimp is a renowned model for the determination of the toxicological impact of substances on living organisms [46]. Doxorubicin served as a control in the current study because it is a well-known chemotherapeutic agent, utilized for the treatment of various cancers types [79]. Thus, the bio-assisted ZnO NPs showed a moderate toxicity, while doxorubicin was considered more toxic in contrast to the NPs because doxorubicin presented an LC50 in the range of 1.0–10.0 µg/mL, while the LC50 of the ZnO NPS was in the range of 10.0–30.0 µg/mL [46,47]. Hemolytic potency is evaluated using different grades, i.e., a material having ≥5% hemolysis is considered hemolytic, while if the hemolytic potency is 2%–5% then it is slightly hemolytic, but if it is ≤2% then it is regarded as non-hemolytic [80]. These findings regarding the hemolytic potential of the ZnO NPs were relevant to previous studies and showed that the ZnO NPs possess slightly hemolytic potency [81].

ROS/RNS are referred to as by-products of the metabolism, which are generated physiologically in the mitochondria. Their production was investigated by utilizing the dihydrorhodamine 123 (DHR123) probe. As the ZnO NPs in this finding increase the level of ROS and RNS in HepG2 cells in contrast to non-treated cells (NTCs), so are our results in accordance with the previous report of [82], suggesting that membrane-integrity is disrupted by an elevated level of the reactive oxygen and nitrogen species. Our results for the ZnO NPs showed decreased mitochondrial integrity, and in the literature, it was indicated that such biosynthesized ZnO NPs have demonstrated a reduced mitochondrial integrity because of metal depletion from their surfaces, which results in an elevated level of ROS that generates oxidation stress as well as membrane-integrity disruption [82]. As protein expression of these caspases was elevated in HepG2 cells, when the NPs were applied in contrast to NTCs, these caspases acted as an effector protein that performs an important role in apoptosis initiation and control over cancer formation [82].

Regarding the antibacterial activity of the ZnO NPs, it has been revealed that significant antibacterial activity against all five of the bacterial strains was shown in our results as a zone of inhibition, measured at ≥12 mm. From the literature, it has been observed that the NPs’ size has a greater impact on their bioactivity, since, due to the smaller size, most of the NPs become accumulated inside the cytoplasm through the cell membrane and, hence, cause more toxicity. The antibacterial activity of the ZnO NPs is also linked with the physiochemical properties of NPs, such as solubility, shape, size, and chemical composition [83]. It is also suggested that the smaller the size an NP is, the greater its efficacy to inhibit bacterial growth [83].

## 5. Conclusions

The present investigation reports the bio-assisted synthesis of the ZnO NPs, with *Lepidium sativum* seed extract as a reducing agent. Characterizations, such as XRD and SEM, were performed to confirm the ZnO NPs’ synthesis. The average particle size of the nanoparticles was evaluated to be 25.6 nm. FTIR analysis revealed the presence of functional groups on the NPs’ surface, and HPLC confirmed the presence of secondary metabolites, i.e., chlorogenic acid, quercetin, and kaempferol, which aid in reduction and capping. Furthermore, the ZnO NPs exhibited potent antioxidant activities, i.e., TAC, TRP, and DPPH, as well as considerable enzyme-inhibition potential for α-amylase, urease, and lipases. Biocompatibility analysis also revealed that these NPs are biocompatible, offer less toxicity to brine-shrimp larvae, and possess low hemolytic potential. The NPs have exhibited dose-dependent toxicity against fibroblast cells, i.e., less toxic at lower doses (25 ug/mL) and more toxic at higher doses (100 µg). Moreover, the ZnO NPs were found to be effectively inhibiting the growth of HepG2 cells. The elevated production of ROS/RNS and enhanced expression of caspase-3/7 was observed in the potential of the ZnO NPs to treat HepG2 cells. Significant antibacterial activity was shown by the NPs against all tested bacterial strains. Therefore, we believe that the bio-assisted ZnO NPs synthesized in the current study might be used for biomedical and pharmaceutical applications, owing to their effective antioxidant, antibacterial, enzyme-inhibition, and anticancer activities. However, for possible biomedical application from the toxicological perspective, further experiments are needed to be completed in animal models, so this study can also lead toward determining the exact mechanism of apoptosis induction via molecular studies as well as facilitate in the future application of the ZnO NPs as anti-cancer therapeutics.

## Figures and Tables

**Figure 1 biomolecules-12-00855-f001:**
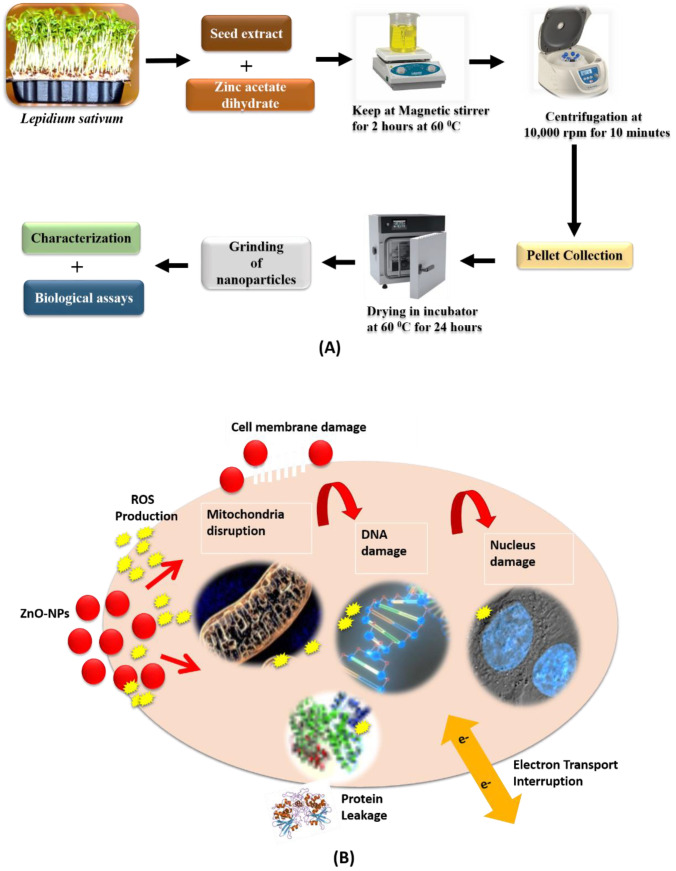
(**A**) Schematic illustration of bio-assisted synthesis of ZnO NPs via green route using *Lepidium sativum* seed extract; (**B**) proposed antibacterial mechanism of action of ZnO NPs. ZnO NPs interact with cell membrane and result in disruption of membrane integrity, thus leading to ion-channel leakage that causes imbalance within the cell. ROS production causes an oxidative stress state in the cell, responsible for protein denaturation, cell-cycle arrest, cellular toxicity, and disruption of mitochondrial function, so the metabolic activities become impaired and finally cell death occurs.

**Figure 2 biomolecules-12-00855-f002:**
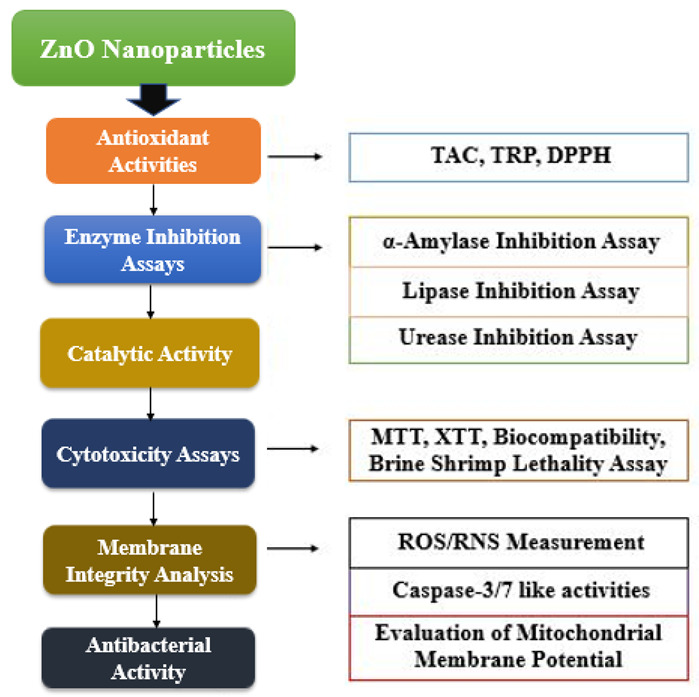
Flow chart of the study design.

**Figure 3 biomolecules-12-00855-f003:**
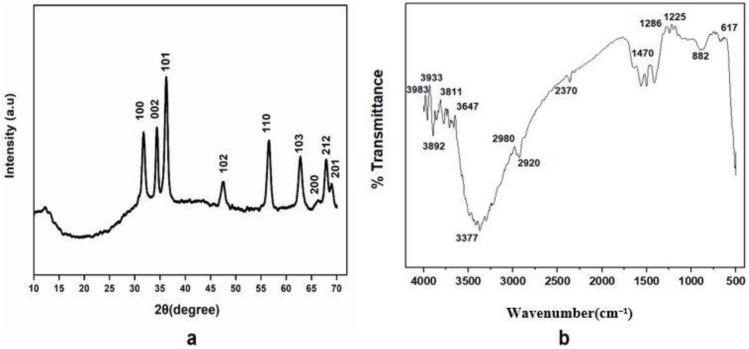
(**a**) XRD of *Lepidium sativum*-mediated synthesis of ZnO NPs; (**b**) FTIR spectra *of Lepidium sativum*-mediated synthesized ZnO NPs.

**Figure 4 biomolecules-12-00855-f004:**
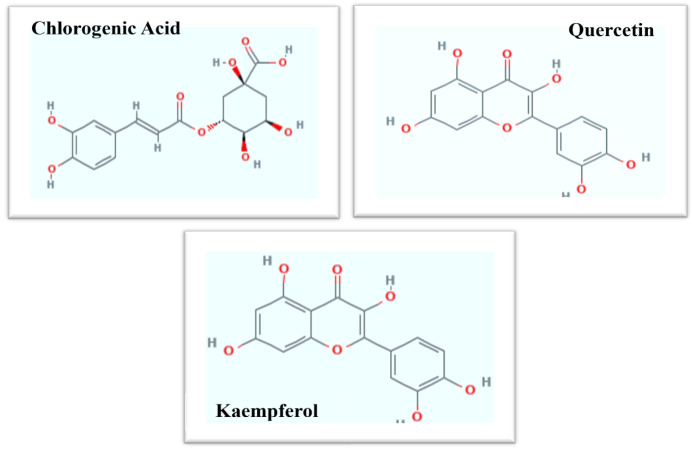
Chemical structure of chlorogenic acid, quercetin, and kaempferol.

**Figure 5 biomolecules-12-00855-f005:**
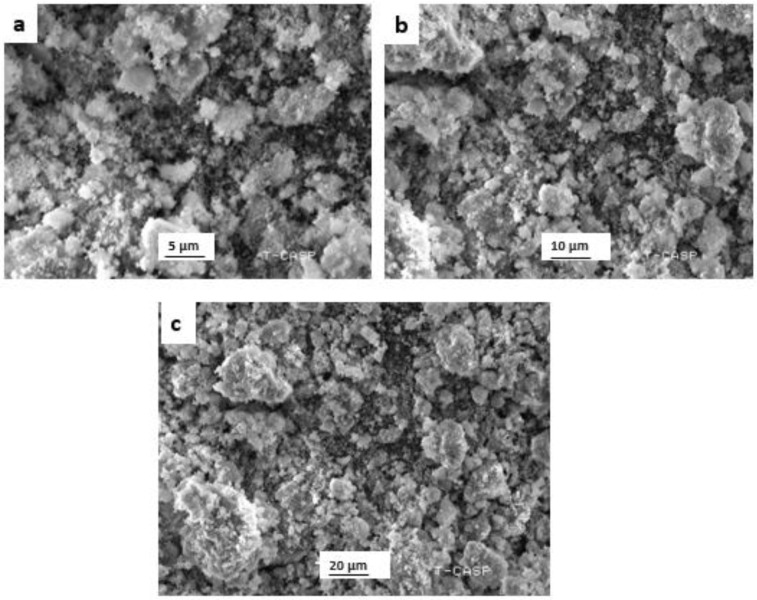
SEM analysis of bio-assisted ZnO NPs at different magnifications (**a**) At 5 μm; (**b**) At 10 μm; (**c**) At 20 μm.

**Figure 6 biomolecules-12-00855-f006:**
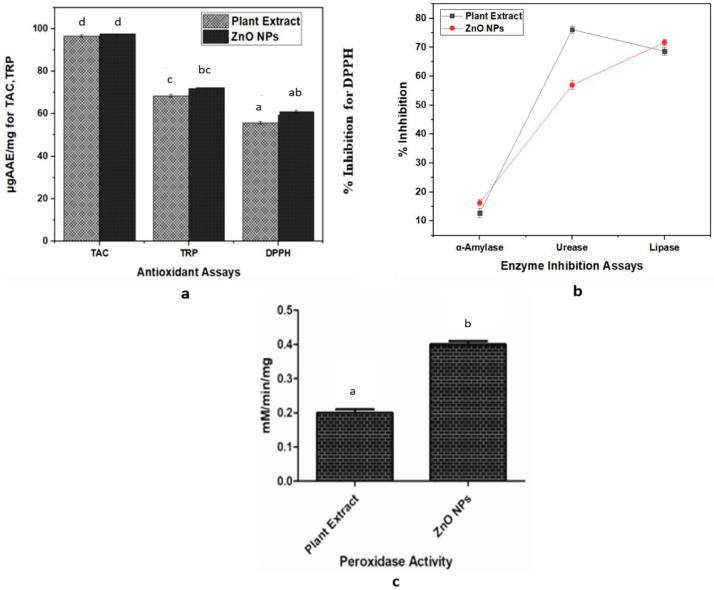
(**a**) Antioxidant assays performed using plant extract and ZnO NPs; (**b**) enzyme-inhibition potential; (**c**) catalytic potential of plant extract and ZnO NPs. Experiments performed in triplicates and values shown as means ± standard deviation. Similar alphabets depicted significant similarity, while differences were shown by different alphabets in the experimental setup (*p* < 0.05).

**Figure 7 biomolecules-12-00855-f007:**
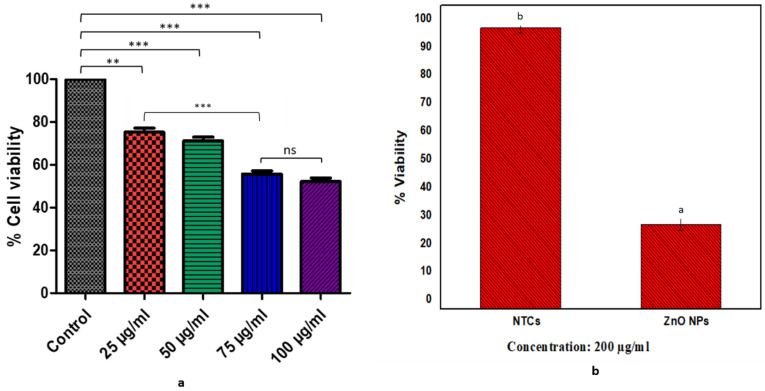
Cytotoxicity assays: (**a**) cell-viability assay (XTT assay) of ZnO NPs against NIH3T3-fibroblast cells; (**b**) anticancer activity (by MTT cell-viability assay) against HepG2 cancer cells. Experiments are performed in triplicates and values are presented as means ± standard deviation. *** results are highly significant, ** significant results, while ns showed that results are not significant; whereas similar alphabets in figure (**b**) illustrated significant similarity, while different alphabets show differences between groups (*p* < 0.05).

**Figure 8 biomolecules-12-00855-f008:**
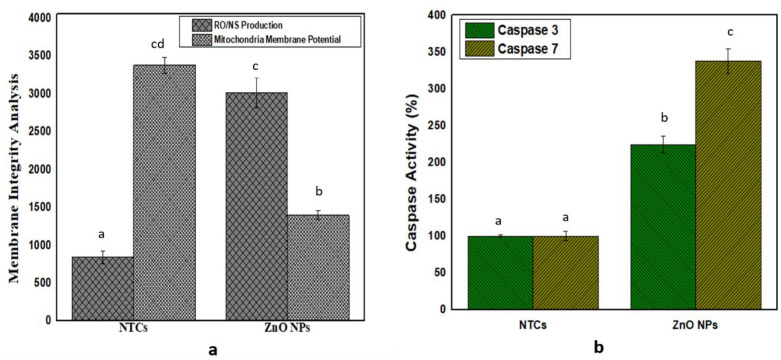
(**a**) Membrane-integrity analysis by ROS/RNS production and evaluation of mitochondrial-membrane potential; (**b**) caspase-3/7 activity experiments were performed in triplicates, and values are shown as means ± standard deviation. Similar alphabets depicted significant similarity, while significant differences are shown by different alphabets in the experimental setup (*p* < 0.05).

**Figure 9 biomolecules-12-00855-f009:**
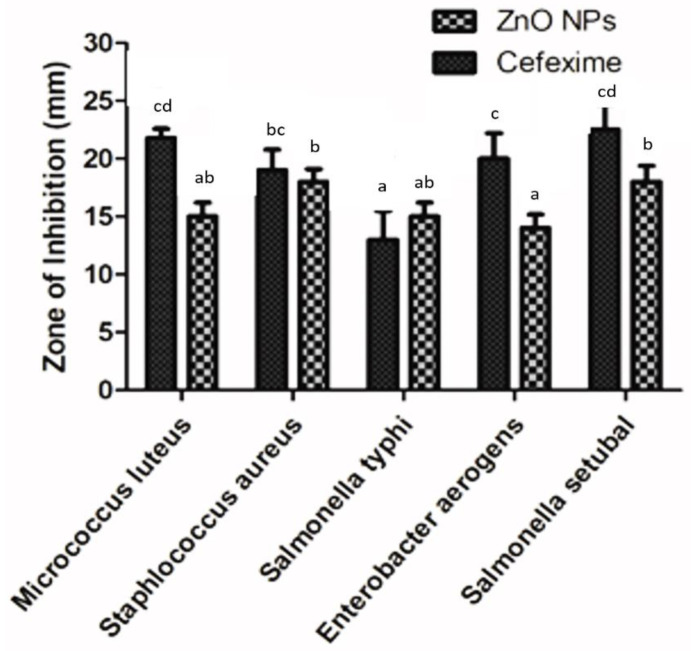
Graphical presentation of antibacterial activity of ZnO NPs. Experiment performed in triplicates, and values shown as means ± standard deviation. Similar alphabets depicted significant similarity, while differences are shown by different alphabets in the experimental setup (*p* < 0.05).

**Table 1 biomolecules-12-00855-t001:** Biocompatibility assays of bio-assisted ZnO NPs.

Assay	Mean ± SD of ZnO NPs	Mean ± SD of Control
Brine-shrimp lethality (LC 50 (in µg/mL)	ZnO NPs 19.4 ± 1.9	Doxorubicin 5.9 ± 0.3
Red blood cells hemolysis (%)	ZnO NPs 4.1 ± 0.2	Non-treated cells NTCs 0.9 ± 0.3

NTCs = non-treated cells, SD = standard deviation.

## Data Availability

All the data supporting the findings of this study are included in this article.

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
