# Peer review of "Bio-Assisted Synthesis and Characterization of Zinc Oxide Nanoparticles from Lepidium sativum and Their Potent Antioxidant, Antibacterial and Anticancer Activities"

_biomolecules, 2022, doi:10.3390/biom12060855_

Round 1

Reviewer 1 Report

Title: “Bio-Assisted Synthesis and Characterization of Zinc Oxide Nanoparticles from Lepidium sativum and their Potent Antioxidant, Antibacterial and Anticancer activities"

Thank you for submitting your article to the Biomolecules.  I have gone through the manuscript; the paper entitled “Bio-Assisted Synthesis and Characterization of Zinc Oxide nanoparticles from Lepidium sativum and their Potent Antioxidant, Antibacterial and Anticancer activities" explains the X-ray structural study, their Potent Antioxidant, Antibacterial and Anticancer activities.

The obtained results are interesting and would be the good contribution in the area of nanoscience and their potential applications. Before publication in Biomolecules, the present manuscript needs few modifications. My concerns are:

1. The abstract should be very concise and organized; it’s too long and should be restructured.

2. You are synthesizing the zinc oxide nanoparticles; using Lepidium sativum seeds, it’s interesting and innovative, could you please explain the role the secondary metabolites in synthesis of zinc oxide nanoparticles; because there are many papers had been successfully reported that using zinc acetate with NaOH solution only.

3. SEM micrograph reveal that nanoparticles were irregular in shape and bears grain boundary, if possible kindly attached EDX with SEM for better understanding of results as well purity of synthesized nanoparticles.

4. You have mentioned in section “2.2. Biosynthesis of ZnO nanoparticles” The supernatant was 120 discarded and the pellet was washed three times with distilled water. The precipitates 121 were re-dissolved in distilled water, poured in the clean petri-plate and kept in an incu-122 bator for drying at 60°C for 24 hrs.  Kindly check the result once again because in most of cases with green method calcinations is required?

5. Here, I would like to suggest you if possible kindly do some characterizations viz TEM, UV-Vis absorption spectroscopy for a better understanding of results as well improving the quality of your manuscript.

6. Kindly cite few more (2-3) references related to the concerned nanoparticles and their applications

Author Response

Response to Reviewers' comments

Reviewer 1:

We sincerely thank the reviewer for his/her thoughtful comments in reviewing our manuscript. We have addressed all the comments in the revised manuscript. The texts have been thoroughly revised. The revisions made during this round of improvement are highlighted in yellow and by track changes which can be seen throughout the manuscript.

Comments and Suggestions for Authors

  1. The abstract should be very concise and organized; it’s too long and should be restructured.

Response: We sincerely thank the reviewer for this comment. The abstract has been thoroughly revised and reconstructed in the revised manuscript.

  1. You are synthesizing the zinc oxide nanoparticles; using Lepidium sativum seeds, it’s interesting and innovative, could you please explain the role the secondary metabolites in synthesis of zinc oxide nanoparticles; because there are many papers had been successfully reported that using zinc acetate with NaOH solution only.

Response: From the literature, it is revealed that many plant extracts were exploited for the synthesis of metal or metal oxide nanoparticles. Synthesis of Nanoparticles entirely relied on the secondary metabolites (phytochemicals). These phytoconstituents are categorized into several classes of compounds such as phenols, amino acids, tannins, carbohydrates, flavonoids, and terpenoids, etc. These phytochemicals play a potent role as reducing agent for the conversion of precursor (metal salts) into metal or metal oxide nanoparticles [1]. It was also demonstrated previously that functional groups in the plant extract donate electrons that are responsible for zinc ions reduction (Zn2+ to Zn+1) and finally to zinc NPs (Zn0) [2]. Thus, in this way secondary metabolites play a role in synthesis of zinc oxide nanoparticles by reducing the precursor Zn2+ into Zn0.

  1. SEM micrograph reveal that nanoparticles were irregular in shape and bears grain boundary, if possible kindly attached EDX with SEM for better understanding of results as well purity of synthesized nanoparticles.

Response: Thank you for your valuable comment. EDX is used for chemical identification of elements and their concentration. In EDX, electrons knock out electrons from atoms, producing X-rays of characteristic wavelength. So that’s why we don’t used EDX because there is no need to determine chemical composition of Zinc oxide nanoparticles. We used alternative technique that is XRD. X-Ray Diffraction (XRD) is used to identify spatial arrangements of atoms in crystalline phases. Moreover, purity of synthesized nanoparticles also determined by XRD [3]. In this study the crystalline nature and purity of NPs was confirmed by the XRD pattern with several diffraction peaks predicted at different 2θ i.e. 31.74⁰, 34.34⁰, 36.35⁰, 47.38⁰, 56.57⁰, 62.68⁰, 66.36⁰, 67.83⁰ and 68.2⁰ corresponding to different miller indices (100), (002), (101), (102), (110), (103), (200), (212) and (201). Through Debye–Scherrer equation the average size of nanoparticle was calculated that was 25.6 nm. Similar results regarding size and shape were also reported in previous studies [4-7].  According to the literature if the size of material falls in between 1-100 nm it is considered as nanostructure [8] .While SEM show the morphology of nanoparticle at different resolution. Previous reported studies also confirmed the nanostructure formation via these techniques as reported by G.Theophil Annand et al. [9] and by Perumal Dhandapani et al [10].

  1. You have mentioned in section “2.2. Biosynthesis of ZnO nanoparticles” The supernatant was 120 discarded and the pellet was washed three times with distilled water. The precipitates 121 were re-dissolved in distilled water, poured in the clean petri-plate and kept in an incu-122 bator for drying at 60°C for 24 hrs. Kindly check the result once again because in most of cases with green method calcinations is required?

Response: Thank you for your valuable comment. I have thoroughly read the methodology. Yes, calcination is required in synthesis of nanoparticles via green method but the protocol followed in this current study is different as in this study ZnO nanoparticles are synthesized by biogenic method (Bio-assisted synthesis). So, calcination is not needed in this study. Moreover, literature study also favoured that calcination is not necessary for synthesis of ZnO nanoparticles. Zaeem, A., Drouet, S., et al. (2020) synthesized biogenic Zinc Oxide Nanoparticles without calcination [11], Bayat, M., Chudinova, E., Zargar, M., et al. (2019) showed a contrast among calcinated and non-calcinated NPs and the non-calcinated NPs also showed confirmation of ZnO NPS synthesis by characterization [12] and Abbasi, B. H., Anjum, S., & Hano, C. (2017) also showed that ZnO NPs synthesized without calcination [13].  In the current study, the evidence of nanostructure of these nanoparticles were provided by X-ray diffraction (XRD) and Scanning electron microscopy (SEM).

  1. Here, I would like to suggest you if possible kindly do some characterizations viz TEM, UV-Vis absorption spectroscopy for a better understanding of results as well improving the quality of your manuscript.

Response: Thank you for your valuable comment. We agreed with your suggestion by adding these two techniques article will become more valuable but in the current study already SEM and XRD were performed to confirm the synthesis of ZnO Nanoparticles and FTIR analysis were also conducted to determine functional groups. Moreover, Debye Scherrer equation also gave average crystalline size i.e. 25.6 nm in the same range as in the literature average crystalline size of ZnO NPs reported  [4-7]. Secondly, our main focus in this current study is to determine the impact of bio-assisted ZnO NPs by antioxidant, enzyme inhibition, catalytic, cytotoxic assays, membrane integrity analysis, and antibacterial activity. So, we focused on in-vitro biological applications so, there is no need to extend characterization by adding more techniques.

  1. Kindly cite few more (2-3) references related to the concerned nanoparticles and their applications

Response: Thank you for your comment. More references related to ZnO nanoparticles and their applications have added and highlighted in yellow in “Introduction Section” in the revised manuscript.

Reviewer 2 Report

Comment

General Overview: The author deals with Bio-Assisted Synthesis and Characterization of Zinc Oxide Nanoparticles from Lepidium sativum and their Potent Antioxidant, Antibacterial and Anticancer activities. The manuscript is good however there are some flaws which need to be modified before it comes to be in publishable form Therefore major revision is recommended. Some of the comments are given below

1-      The abstract should be precisely written. Which should include objective, methodology, as well as novelty of the study

2-      In the introduction please include some more studies as the present form is very simple.

3-      Please include flowchart to describe methodology

4-      Improve DPI of figure 5b

5-      Conclusion must be precisely written. Please include recommendation for future research.

Author Response

Reviewer 2:

Comments and Suggestions for Authors

  • The abstract should be precisely written. Which should include objective, methodology, as well as novelty of the study

Response: Thank you for your valuable comment. The abstract has been thoroughly revised, reconstructed and improved in the revised manuscript. Moreover, as you suggested objective, methodology and novelty of the study also added in the abstract.

  • In the introduction please include some more studies as the present form is very simple.

Response: We would sincerely thanks for this comment. The Introduction has been thoroughly revised and some more studies have been added along with references in the lines 80-83, 97-101, and 103-106 in the Introduction section in the revised manuscript.

  • Please include flowchart to describe methodology

Response: We would like to thank reviewer for this comment. The flow chart of methodology has been added in the manuscript file by the name “Figure.2”.

  • Improve DPI of figure 5b

Response: Thank you for your valuable comment. DPI of Figure 5b has been improved and added in the revised manuscript and named as Figure 6b because of addition of flow chart figure numbering changed. Now the image is clearly visible.

5-      Conclusion must be precisely written. Please include recommendation for future research.

Response: We would like to thank for this comment. Conclusion has been revised and written precisely. Moreover, future perspective has also been added in the revised manuscript.

Reviewer 3 Report

Dear Authors, in your interesting manuscript, the following points should be added/changed to further improve it:

1.       Abstract: I have a comment on the sentence “XRD and SEM revealed morphologically hexagonal nanoparticles in the range of 25.6 nm while FTIR analysis and HPLC predicted the functional groups.” Please specify which particle size you mean (e.g. average particle size).

2.       Graphic abstract : Please explain to me what the word “reduction” refers to in the synthesis of ZnO (under “ZnO Nanoparticles Formation).

3.       Introduction: I have a comment on the sentence “ZnO NPs prepared by conventional techniques like laser ablation [11], sol-gel method, solvothermal, chemical reduction [12] and inert gas condensation.” I suggest mentioning the microwave method for obtaining nano ZnO and referring to relevant review articles (e.g. DOI: DOI:10.3390/nano10061086, DOI:10.1080/10408436.2021.1886041). lease explain to me what the authors mean by the name of the method "chemical reduction”? Which chemical reduction is tum referred to in the preparation of ZnO?

4.       Introduction: I have a comment on the sentence “The current study aimed to synthesize bio-assisted ZnO NPs from seeds of Lepidium 79 sativum. Genus Lepidium belongs to the family of Brassicaceae, it is basically an edible 80 herb with length of approximately 50 cm.” Please highlight/indicate the novelty of your work.

5.       Biosynthesis of ZnO nanoparticles: Please add information on the reagents used (purity, manufacturer)

6.       Biosynthesis of ZnO nanoparticles: Please explain to me why the authors claim to have obtained ZnO using biosynthesis?

7.       Biosynthesis of ZnO nanoparticles: I have a comment on the sentence “Medicinal plant Lepidium sativum. was exploited for the successful biosynthesis of ZnO NPs for the very first time.” How does this statement by the authors relate to the article “Phytotoxic activity of the zinc oxyde nanoparticles synthesized from different precursors on germination and radicle growth of seeds lepidium sativum” International Journal of Scientific and Research Publications, Volume 4, Issue 12, December 2014

8.       X-Ray Diffraction (XRD) Analysis: I have a comment on the sentence “The crystalline size of ZnO NPs was obtained by Debye-Sherrer equation.” The authors have given a wrong name to the equation, please correct the name of the equation (DOI:10.1038/nnano.2011.145).

9.       Materials and Methods: Please add information on the analysers used (model, manufacturer).

10.   XRD (X-Ray Diffraction) Analysis: Please refer to the agreement with the XRD results for the reference sample (ZnO JCPDS 36-1451).

11.   Scanning Electron Microscopy (SEM) Analysis: Where is the evidence (results) that the authors obtained ZnO nanoparticles? How can the reader, based on SEM images from scale bar 5um and 10um, confirm that nanoparticles were obtained? Please add the SEM results that confirm that ZnO nanoparticles were obtained.

Author Response

Reviewer 3:

Dear Authors, in your interesting manuscript, the following points should be added/changed to further improve it:

  1. Abstract: I have a comment on the sentence “XRD and SEM revealed morphologically hexagonal nanoparticles in the range of 25.6 nm while FTIR analysis and HPLC predicted the functional groups.” Please specify which particle size you mean (e.g. average particle size)

Response: We would like to thank you for this comment. This line is changed in the abstract in the revised manuscript and also given here “Crystalline, hexagonal structured NPs with an average crystalline size distribution of 25.6 nm was obtained.” As it is specified to add average particle size the word average has been added in the revised manuscript.

  1. Graphic abstract : Please explain to me what the word “reduction” refers to in the synthesis of ZnO (under “ZnO Nanoparticles Formation).

Response: In the current study Lepidium sativum seed extract utilized as capping and reducing agent because the extract has the phytochemicals present in it that reduced the precursor Zn2+ into Zn0. Moreover, the presence of phytochemicals like molecules attached on the surface of nanoparticles were also confirmed by FTIR and HPLC. By deeply studying the phytochemical constituents of the chosen plant, seed extract was used to reduce the precursor Zn2+.  Antioxidant results of the current study also revealed that some of the compounds of the aqueous extract of Lepidium sativum responsible in the stabilization as well as reduction of ZnO NPs during the synthesis of NPs. From the literature it is also revealed that functional groups in the plant extract donate electrons that are responsible for zinc ions reduction (Zn2+ to Zn+1) and finally to zinc NPs (Zn0) [2]. This is referred as reduction.

  1. Introduction: I have a comment on the sentence “ZnO NPs prepared by conventional techniques like laser ablation [11], sol-gel method, solvothermal, chemical reduction [12] and inert gas condensation.” I suggest mentioning the microwave method for obtaining nano ZnO and referring to relevant review articles (e.g. DOI: DOI:10.3390/nano10061086, DOI:10.1080/10408436.2021.1886041). lease explain to me what the authors mean by the name of the method "chemical reduction”? Which chemical reduction is tum referred to in the preparation of ZnO?

Response:  We would like to thank for this comment. Microwave method has been added and these two references that are mentioned in this comment has also been cited and highlighted in revised manuscript. Moreover, from the literature studies it is illustrated that chemical reduction method is utilized to synthesize spherical ZnO nanocrystals by exploiting different metal-organic precursors e.g. (zinc(II)-1,3,5-benzenetricarboxylate hydrate and zinc(II)-1,2,4,5-benzenetetracarboxylate hydrate via novel chemical reduction method. Moreover, triethylamine (Et3N) also utilized in contrast to conventional hydroxide bases. Because it is mild base than hydroxide and forms hydroxide ions via hydrolysis in water and thus modulated OHconcentration in the medium and thus avoid the production of Zn(OH)2 and resulted in synthesis of ZnO NPs [14].

  1. Introduction: I have a comment on the sentence “The current study aimed to synthesize bio-assisted ZnO NPs from seeds of Lepidium 79 sativum. Genus Lepidium belongs to the family of Brassicaceae, it is basically an edible 80 herb with length of approximately 50 cm.” Please highlight/indicate the novelty of your work.

Response: We would sincerely thank you for this comment. The novelty of the work has also been added and highlighted yellow in the introduction section (last paragraph) in the revised manuscript and also illustrated here:

  1. This is the first ever study in which zinc oxide nanoparticles were synthesized from seeds extract of Lepidium sativum.
  2. Normally Salmonella typhi showed resistant against antibiotics but in the current study significant antibacterial activity was examined in contrast to most potent antibiotic ‘cefexime”.
  3. Furthermore, the current study is focussed on in-vitro biological applications so, performed several biological assays such as antioxidant activities, enzyme inhibition assays, catalytic activity, cytotoxicity assays, membrane integrity analysis and antibacterial activity.
  4. Biosynthesis of ZnO nanoparticles: Please add information on the reagents used (purity, manufacturer)

Response: Thank you for your valuable comment. Reagents used in the current study has been added and highlighted yellow in the Methodology section by the name 2.1. “Chemicals”.

  1. Biosynthesis of ZnO nanoparticles: Please explain to me why the authors claim to have obtained ZnO using biosynthesis?

Response: Thank you for your valuable comment. In contrast to chemical approach, ZnO nanoparticles are preferred to synthesize by biological method. Because green synthesis has several benefits over chemical synthesis like this process of synthesis is cost effective, simple, no harmful waste products, ecofriendly and no need of additional capping agents as well as surfactants as metabolites of plants perform a role as capping and stabilizing agent [15].   For the synthesis of ZnO NPs, plant is best and appropriate for the large-scale production of nanoparticles than other organisms. NPs derived from plants are more diverse and stable in size and shape in contrast to synthesized by other organisms [16]. Thus, biosynthesis is the most appropriate method for synthesis of ZnO NPs.

  1. Biosynthesis of ZnO nanoparticles: I have a comment on the sentence “Medicinal plant Lepidium sativum. was exploited for the successful biosynthesis of ZnO NPs for the very first time.” How does this statement by the authors relate to the article “Phytotoxic activity of the zinc oxyde nanoparticles synthesized from different precursors on germination and radicle growth of seeds lepidium sativum” International Journal of Scientific and Research Publications, Volume 4, Issue 12, December 2014

Response: Thank you for your valuable comment. In the current study, ZnO NPs were synthesized by Lepidium sativum and its in-vitro biological applications were predicted but the article which you mentioned in this comment tells about the impact of ZnO NPs on the germination and radical growth of seeds. These articles related in this way that in current study and in the published article precursor is same Zinc acetate dihydrate but two more precursor utilized in published article. Moreover, in current study and published article both exploited the seeds of Lepidium sativum but the methodology is different. Current study utilized seed extract for synthesis of ZnO NPs while the published article utilized seeds for the purpose to determine phytotoxic impact of nanoparticles on plants. In both studies ZnO NPs synthesized but the method is different as current study utilized bio-assisted synthesis (green synthesis) while the published article synthesized nanoparticles by chemical method. The last but not least thing that related our study with that published article is that as the demand to enhance crop productivity and resistance directs interest in nanotechnology. So, we in future will also enable to check our synthesized nanoparticles impact on the Lepidium sativum as the published article did but in our study design the source of synthesis of ZnO NPs will be same with that on which we will want to predict phytotoxic impact of nanoparticles but in the published article nanoparticles synthesized chemically and then determine phytotoxic impact on plant.

  1. X-Ray Diffraction (XRD) Analysis: I have a comment on the sentence “The crystalline size of ZnO NPs was obtained by Debye-Sherrer equation.” The authors have given a wrong name to the equation, please correct the name of the equation (DOI:10.1038/nnano.2011.145).

Response: We would like to thank Reviewer for this comment. We have corrected the mentioned spelling mistakes and have thoroughly revised the manuscript.

  1. Materials and Methods: Please add information on the analysers used (model, manufacturer).

Response: Thank you for your valuable comment. Information on the analysers used have been added in the revised manuscript in Materials and Methods section. Here these are also given below:

X-Ray Diffractometer (AXS DS Advance, Bruker, Billerica, MA, USA) ; Fourier Transform Infrared Radiation Spectroscopy (FTIR, Bruker, Billerica, MA, USA) ; HPLC (Merck, Saint Quentin Fallavier, France) analysis ; Scanning electron microscope (SEM, Jeol JSM-6510LV) ; Centrifuge (Spectrafuge™ 24D microcentrifuge, Labnet International, Corning, NY, USA) ; Microplate reader (Thermo Scientific Multiskan GO) ; UV spectrophotometer (Halo DR-20, UV-VIS spectrophotometer, Dynamica Ltd, Victoria, Australia) and Origin 8.5 (Windows v8.1, Northampton, MA, USA).

  1. XRD (X-Ray Diffraction) Analysis: Please refer to the agreement with the XRD results for the reference sample (ZnO JCPDS 36-1451).

Response: Thank you for your valuable comment. ZnO NPs in the current study were determined by X-ray diffraction. Confirmation of crystalline nature and purity of ZnO NPs were done by XRD pattern with strong diffraction peaks analyzed at different 2θ i.e. 31.74⁰, 34.34⁰, 36.35⁰, 47.38⁰, 56.57⁰, 62.68⁰, 66.36⁰, 67.83⁰ and 68.2⁰ corresponding to different miller indices (100), (002), (101), (102), (110), (103), (200), (212) and (201) respectively. It is revealed that diffraction peaks and miller indices of our ZnO NPs are in accordance with the previous studies i.e. Arakha, M., et al., (2015) [4], Matinise, N., et al.,(2017) [5], Vijayakumar, S., et al.,(2016) [6], Janaki, A.C., et al., (2015) [7]. Moreover, literature studies revealed that upon the confirmation of synthesis of ZnO NPs by XRD they mentioned ZnO JCPDS 36-1451 So, in the current study we also mentioned that for confirmation but to avoid ambiguity citation of this reference sample (ZnO JCPDS 36-1451) has also been added in the revised manuscript and highlighted in yellow which means that our results are in accordance with Krishna Reddy, G.,et al., (2017) [17].

  1. Scanning Electron Microscopy (SEM) Analysis: Where is the evidence (results) that the authors obtained ZnO nanoparticles? How can the reader, based on SEM images from scale bar 5um and 10um, confirm that nanoparticles were obtained? Please add the SEM results that confirm that ZnO nanoparticles were obtained.

Response: Thank you for your valuable comment. SEM images were taken at different magnifications i.e. 5um and 10um, from the pictures and previous reported studies, the morphology of nanoparticles was confirmed. Moreover, the average size of nanoparticles was calculated through ImageJ software i.e. 30nm which is in accordance with the average particle size obtained by XRD (25.6nm). Although the magnifications is low but sufficient to calculate the size through this software. Previous reported studies also confirmed the nanostructure formation via these techniques as reported by G.Theophil Annand et al. [9] and by Perumal Dhandapani et al [10].

References:

  1. Madhumitha, G., G. Elango, and S.M. Roopan, Biotechnological aspects of ZnO nanoparticles: overview on synthesis and its applications. Applied microbiology and biotechnology, 2016. 100(2): p. 571-581.
  2. Alamdari, S., et al., Preparation and Characterization of Zinc Oxide Nanoparticles Using Leaf Extract of Sambucus ebulus. Applied Sciences, 2020. 10(10): p. 3620.
  3. Monsef, R., M. Ghiyasiyan-Arani, and M. Salavati-Niasari, Application of ultrasound-aided method for the synthesis of NdVO4 nano-photocatalyst and investigation of eliminate dye in contaminant water. Ultrasonics Sonochemistry, 2018. 42: p. 201-211.
  4. Arakha, M., et al., The effects of interfacial potential on antimicrobial propensity of ZnO nanoparticle. Scientific reports, 2015. 5(1): p. 1-10.
  5. Matinise, N., et al., ZnO nanoparticles via Moringa oleifera green synthesis: Physical properties & mechanism of formation. Applied Surface Science, 2017. 406: p. 339-347.
  6. Vijayakumar, S., et al., Laurus nobilis leaf extract mediated green synthesis of ZnO nanoparticles: characterization and biomedical applications. Biomedicine & Pharmacotherapy, 2016. 84: p. 1213-1222.
  7. Janaki, A.C., E. Sailatha, and S. Gunasekaran, Synthesis, characteristics and antimicrobial activity of ZnO nanoparticles. Spectrochimica Acta Part A: Molecular and Biomolecular Spectroscopy, 2015. 144: p. 17-22.
  8. Kaliraj, L., et al., Synthesis of panos extract mediated ZnO nano-flowers as photocatalyst for industrial dye degradation by UV illumination. Journal of Photochemistry and Photobiology B: Biology, 2019. 199: p. 111588.
  9. Anand, G.T., et al., Green synthesis of ZnO nanoparticle using Prunus dulcis (Almond Gum) for antimicrobial and supercapacitor applications. Surfaces and Interfaces, 2019. 17: p. 100376.
  10. Dhandapani, P., et al., Ureolytic bacteria mediated synthesis of hairy ZnO nanostructure as photocatalyst for decolorization of dyes. Materials Chemistry and Physics, 2020. 243: p. 122619.
  11. Zaeem, A., et al., Effects of biogenic zinc oxide nanoparticles on growth and oxidative stress response in flax seedlings vs. in vitro cultures: a comparative analysis. Biomolecules, 2020. 10(6): p. 918.
  12. Bayat, M., et al., Phyto-assisted green synthesis of zinc oxide nanoparticles and its antibacterial and antifungal activity. Research on Crops, 2019. 20(4): p. 725-730.
  13. Abbasi, B.H., S. Anjum, and C. Hano, Differential effects of in vitro cultures of Linum usitatissimum L.(Flax) on biosynthesis, stability, antibacterial and antileishmanial activities of zinc oxide nanoparticles: a mechanistic approach. RSC advances, 2017. 7(26): p. 15931-15943.
  14. Shit, S., T. Kamilya, and P.K. Samanta, A novel chemical reduction method of growing ZnO nanocrystals and their optical property. Materials Letters, 2014. 118: p. 123-125.
  15. Nazir, S., et al., Synthesis, characterisation and bactericidal effect of ZnO nanoparticles via chemical and bio-assisted (Silybum marianum in vitro plantlets and callus extract) methods: a comparative study. IET nanobiotechnology, 2018. 12(5): p. 604-608.
  16. Ramesh, P., A. Rajendran, and M. Meenakshisundaram, Green syntheis of zinc oxide nanoparticles using flower extract cassia auriculata. Journal of NanoScience and NanoTechnology, 2014. 2(1): p. 41-45.
  17. Krishna Reddy, G., et al., Luminescence and spectroscopic investigations on Gd3+ doped ZnO nanophosphor. Journal of Asian Ceramic Societies, 2017. 5(3): p. 350-356.

Round 2

Reviewer 2 Report

The authors have modified the manuscript as recommended.

Author Response

Thank you very much for your positive reply.

Reviewer 3 Report

1.      Graphic abstract : Please explain to me what the word “reduction” refers to in the synthesis of ZnO (under “ZnO Nanoparticles Formation).

Response: In the current study Lepidium sativum seed extract utilized as capping and reducing agent because the extract has the phytochemicals present in it that reduced the precursor Zn2+ into Zn0. Moreover, the presence of phytochemicals like molecules attached on the surface of nanoparticles were also confirmed by FTIR and HPLC. By deeply studying the phytochemical constituents of the chosen plant, seed extract was used to reduce the precursor Zn2+.  Antioxidant results of the current study also revealed that some of the compounds of the aqueous extract of Lepidium sativum responsible in the stabilization as well as reduction of ZnO NPs during the synthesis of NPs. From the literature it is also revealed that functional groups in the plant extract donate electrons that are responsible for zinc ions reduction (Zn2+ to Zn+1) and finally to zinc NPs (Zn0) [2]. This is referred as reduction.

Reviewer's reply: I do not understand the point of the authors' response. I point out to the authors that they obtained zinc oxide (ZnO) and not zinc (Zn) metal. So where is the evidence here that the reduction reaction took place? Please add the chemical reaction equation for obtaining ZnO for the reagents used.

2.      Introduction: I have a comment on the sentence “ZnO NPs prepared by conventional techniques like laser ablation [11], sol-gel method, solvothermal, chemical reduction [12] and inert gas condensation.” I suggest mentioning the microwave method for obtaining nano ZnO and referring to relevant review articles (e.g. DOI: DOI:10.3390/nano10061086, DOI:10.1080/10408436.2021.1886041). lease explain to me what the authors mean by the name of the method "chemical reduction”? Which chemical reduction is tum referred to in the preparation of ZnO?

Response:  We would like to thank for this comment. Microwave method has been added and these two references that are mentioned in this comment has also been cited and highlighted in revised manuscript. Moreover, from the literature studies it is illustrated that chemical reduction method is utilized to synthesize spherical ZnO nanocrystals by exploiting different metal-organic precursors e.g. (zinc(II)-1,3,5-benzenetricarboxylate hydrate and zinc(II)-1,2,4,5-benzenetetracarboxylate hydrate via novel chemical reduction method. Moreover, triethylamine (Et3N) also utilized in contrast to conventional hydroxide bases. Because it is mild base than hydroxide and forms hydroxide ions via hydrolysis in water and thus modulated OHconcentration in the medium and thus avoid the production of Zn(OH)2 and resulted in synthesis of ZnO NPs [14].

Reviewer's reply: Please do not reproduce errors. I repeat my comment on the sentence “ZnO NPs prepared by conventional techniques like laser ablation 113 [15], sol-gel method, solvothermal, chemical reduction [16], microwave method [17, 18], and inert gas condensation.” Please give me an example of a chemical reduction reaction to obtain ZnO. What does reference number 16 with the title " Synthesis of silver nanoparticles by chemical reduction method and their antibacterial activity" have to do with obtaining ZnO by chemical reduction?

3.      Biosynthesis of ZnO nanoparticles: Please explain to me why the authors claim to have obtained ZnO using biosynthesis?

Response: Thank you for your valuable comment. In contrast to chemical approach, ZnO nanoparticles are preferred to synthesize by biological method. Because green synthesis has several benefits over chemical synthesis like this process of synthesis is cost effective, simple, no harmful waste products, ecofriendly and no need of additional capping agents as well as surfactants as metabolites of plants perform a role as capping and stabilizing agent [15].   For the synthesis of ZnO NPs, plant is best and appropriate for the large-scale production of nanoparticles than other organisms. NPs derived from plants are more diverse and stable in size and shape in contrast to synthesized by other organisms [16]. Thus, biosynthesis is the most appropriate method for synthesis of ZnO NPs.

Reviewer's reply:

To clarify definitions:

Green synthesis of nanomaterials refers to the synthesis of different nanoparticles using bioactive agents such as plant materials, microorganisms, and various biowastes including vegetable waste, fruit peel waste, eggshell, agricultural waste, and so on.

biosynthesis - the process of producing a particular substance within a living organism.

For biogenic synthesis, simply biological extracts are mixed with the metal salt solutions and the effect of different parameters such as the concentration of the metal salt and extract, pH, temperature, time and radiation are studied.

Please use the correct definitions in your work.

4        Scanning Electron Microscopy (SEM) Analysis: Where is the evidence (results) that the authors obtained ZnO nanoparticles? How can the reader, based on SEM images from scale bar 5um and 10um, confirm that nanoparticles were obtained? Please add the SEM results that confirm that ZnO nanoparticles were obtained.

Response: Thank you for your valuable comment. SEM images were taken at different magnifications i.e. 5um and 10um, from the pictures and previous reported studies, the morphology of nanoparticles was confirmed. Moreover, the average size of nanoparticles was calculated through ImageJ software i.e. 30nm which is in accordance with the average particle size obtained by XRD (25.6nm). Although the magnifications is low but sufficient to calculate the size through this software. Previous reported studies also confirmed the nanostructure formation via these techniques as reported by G.Theophil Annand et al. [9] and by Perumal Dhandapani et al [10].

Reviewer's reply: It does not accept the authors' explanations. I reiterate my request to add SEM results confirming receipt of nanoparticles. The reader is unable to confirm from Figure 5 that 30nm nanoparticles have been obtained. Crystallite size versus particle size are different definitions. Please add evidence that the nanoparticles obtained consist of single crystallites.

5.      X-Ray Diffraction (XRD) Analysis: I have a comment on the sentence “The crystalline size of ZnO NPs was obtained by Debye-Sherrer equation.” The authors have given a wrong name to the equation, please correct the name of the equation (DOI:10.1038/nnano.2011.145).

Response: We would like to thank Reviewer for this comment. We have corrected the mentioned spelling mistakes and have thoroughly revised the manuscript.

Reviewer's reply: The first scientist to investigate the effect of limited particle size on X-ray diffraction patterns was Paul Scherrer, who published his results in a paper that included what became known as the Scherrer equation. However, it seems to us that this equation is often erroneously referred to as the 'Debye–Scherrer equation'. Indeed, strictly speaking, there is no Debye-Scherrer equation. The correct name for the equation is Scherrer equation.

Author Response

see attached file (responses in blue)
